# Experimental limit on an exotic parity-odd spin- and velocity-dependent interaction using an optically polarized vapor

Young Jin Kim [1], Ping-Han Chu[1], Igor Savukov[1] & Shaun Newman[1]

Exotic spin-dependent interactions between fermions have recently attracted attention in relation to theories beyond the Standard Model. The exotic interactions can be mediated by hypothetical fundamental bosons which may explain several unsolved mysteries in physics. Here we expand this area of research by probing an exotic parity-odd spin- and velocity-dependent interaction between the axial-vector electron coupling and the vector nucleon coupling for polarized electrons. This experiment utilizes a high-sensitivity atomic magnetometer, based on an optically polarized vapor that is a source of polarized electrons, and a solid-state mass containing unpolarized nucleons. The atomic magnetometer can detect an effective magnetic field induced by the exotic interaction between unpolarized nucleons and polarized electrons. We set an experimental limit on the electron-nucleon coupling $g_A^e g_V^N < 10^{-30}$ at the mediator boson mass below $10^{-4}$ eV, significantly improving the current limit by up to 17 orders of magnitude.

---

[1] P-21, Los Alamos National Laboratory, P.O. Box 1663MS-D454, Los Alamos, NM 87545, USA. Correspondence and requests for materials should be addressed to Y.J.K. (email: youngjin@lanl.gov) or to P.-H.C. (email: pchu@lanl.gov)

The searches for the exotic spin-dependent interactions between fermions[1] have recently attracted attention[2]. To date, 15 possible interactions have been derived by considering the spins and the relative velocity of two interacting fermions[1,3]. In recent theoretical literature, mediators of such interactions have been generically defined as WISPs (weakly interacting sub-eV particles)[4] including the most plausible mediators, the axion and the hidden photon. The axion is a hypothetical spin-0 pseudoscalar boson that can explain the puzzling strong charge-parity (CP) problem in the quantum chromodynamics (QCD)[5–7] and is a theoretically well-motivated candidate for dark matter in the Universe[8,9]. The hidden photon[10–12] is a hypothetical spin-1 boson arising from the breaking of a U(1) gauge symmetry, which can be a dark matter candidate as well, indirectly interacting with ordinary particles[13,14]. Detection of WISPs is challenging because they are only very weakly coupled to fermions to mediate interactions between the fermions. Recent WISPs searches, such as the Axion Dark Matter eXperiment (ADMX)[15], are reviewed in ref. [16].

Searches for exotic interactions have been typically conducted by closely positioning two objects and using a sensitive detector such as a superconducting quantum interference device (SQUID)[17], electron-spin polarized torsion-pendulum[18], or precessing nuclear spins in nuclear magnetic resonance (NMR)[19]. Recently, we have expanded this area of research using a highly sensitive non-cryogenic atomic magnetometer based on an optically pumped alkali-metal atomic vapor[20,21]. The atomic magnetometer operates in the spin-exchange relaxation-free (SERF) regime in which the effects of spin-exchange collisions on spin relaxation are tuned off, dramatically improving the sensitivity to the femto-Tesla level[22,23]. Our approach probes exotic interactions between optically polarized electrons in an atomic vapor and unpolarized or polarized particles of a solid-state mass in the vicinity of the atomic vapor.

Since the first parity-odd interaction has been discovered in the weak interaction sector[24,25], various searches for parity-odd interactions have been performed, for example, atomic parity non-conservation (PNC) experiments[26] and electric dipole moment (EDM) experiments[27,28]. The searches could be pivotal for developing theories beyond the Standard Model of particle physics. In the present study we investigate an exotic parity-odd spin- and velocity-dependent interaction of electrons with axial-vector electron coupling and vector nucleon coupling ($g_A^e g_V^N$)[3], where A, V, e, and N stand for the axial-vector coupling, the vector coupling, the electron, and the nucleon, respectively. This interaction can be generated by massive spin-1 boson exchange[29], for example, from spontaneous breaking with two or more Higgs doublets. Some experiments on the interaction of neutrons have been conducted using polarized neutrons in liquid $^4$He[30], combining experimental constraints from other experiments[31] and polarized $^3$He coupling to the Earth[32]. While the parity-even spin- and velocity-dependent interaction for electrons has been experimentally investigated[20,33–36], to the best of our knowledge, the parity-odd spin- and velocity-dependent interaction of electrons has been explored by few laboratory experiments such as electron-spin polarized torsion-pendulum experiments[33,34] using the Sun and the Moon as a unpolarized test mass.

In the following, we explore the parity-odd spin- and velocity-dependent interaction of electrons using a SERF atomic magnetometer based on a similar experimental approach in refs. [20,21]. The experiment sets a limit on the electron-nucleon coupling $g_A^e g_V^N < 10^{-30}$ for the boson mass less than $10^{-4}$ eV, equivalent to the interaction range larger than $10^{-3}$ m. This improves the current limit by up to 17 orders of magnitude.

## Results

**Exotic parity-odd spin- and velocity-dependent interaction.** We investigated the interaction between polarized electrons and unpolarized nucleons based on a SERF atomic magnetometer:

$$V(\hat{\boldsymbol{\sigma}}_i, \mathbf{r}) = g_A^e g_V^N \frac{\hbar}{8\pi}(2\hat{\boldsymbol{\sigma}}_i \cdot \mathbf{v})\frac{e^{-r/\lambda}}{r}, \quad (1)$$

where $\hbar$ is Planck's constant, $\hat{\boldsymbol{\sigma}}_i$ is the $i$th spin unit vector of the polarized electron, $\mathbf{r}$ is the separation vector in the direction between the polarized electron and the unpolarized nucleon, $r = |\mathbf{r}|$, $\mathbf{v}$ is their relative velocity vector, and $\lambda = \hbar(m_b c)^{-1}$ is the interaction range (or the boson Compton wavelength) with $m_b$ being the boson mass and $c$ being the speed of light in vacuum. $g_A^e g_V^N$ is the interaction coupling assuming no difference between electron-neutron and electron-proton couplings, and ignoring electron-electron couplings. The interaction induces the energy shift $\Delta E$ of the polarized electrons of the alkali-metal atoms in a SERF atomic vapor, which can be recast as

$$V = \Delta E = \gamma\hbar\hat{\boldsymbol{\sigma}}_i \cdot \mathbf{B}_{\mathrm{eff}}, \quad (2)$$

where $\gamma$ is the gyromagnetic ratio of the alkali-metal atoms and $\mathbf{B}_{\mathrm{eff}}$ is an effective magnetic field at the location of the atomic vapor, produced by the interaction $V$. From Eqs. (1) and (2), the effective magnetic field is

$$\mathbf{B}_{\mathrm{eff}} = g_A^e g_V^N \frac{2\mathbf{v}}{8\pi\gamma}\frac{e^{-r/\lambda}}{r}. \quad (3)$$

This experiment is to detect the $\mathbf{B}_{\mathrm{eff}}$ with a SERF atomic magnetometer.

The SERF atomic magnetometer contains two laser beams (for detail, see for example the description of a SERF magnetometer in ref. [37]). The circularly polarized pump laser beam serves to pump atomic spins into the stretched state, creating nearly 100% polarization. The linearly polarized probe laser beam serves to detect the spin projection along the probe beam. The non-zero spin projection results in the rotation of light polarization, due to the strong birefringence of the spin-polarized vapor, and is measured with a polarizing beam splitter and two photo detectors as a small difference in the balanced output. The interaction of an external magnetic field with the polarized atomic spins leads to a change in the orientations of the spins and hence in the observed light polarization rotation. Because Eq. (2) formally describes the interaction of electron spins with the effective magnetic field, the interaction $V$ causes similar changes in the spin orientation as the ordinary field, and can be probed with the magnetometer[20,21].

**Experimental details.** We employed a commercial cm-scale SERF magnetometer which is low cost, compact, and easy to operate[38,39]. It is based on a rubidium (Rb) $3 \times 3 \times 3$ mm$^3$ atomic vapor cell with a buffer gas. In order to operate the magnetometer in the SERF regime, the Rb vapor was heated to ~160 °C, which supplies sufficiently large Rb density (~$10^{13}$ Rb atoms) as the source of optically polarized electrons. Furthermore, the magnetometer was placed inside a ferrite cylindrical shield (OD = 18 cm, height $H = 38$ cm, material thickness $t = 0.6$ cm) with end-caps and a three-layer open $\mu$-metal co-axial cylinder (ID = 23 cm, OD = 29 cm, outer $H = 69$ cm) as shown in Fig. 1a. The magnetic shields sufficiently suppressed the external dc magnetic field and magnetic noise, but the residual fields inside the ferrite shield were compensated with three orthogonal coils [not shown in Fig. 1a]. The magnetometer was rigidly mounted inside the ferrite shield by a plastic holder.

For the source of unpolarized nucleons, we used a $2 \times 2 \times 2$ cm$^3$ non-magnetic bismuth germanate insulator (BGO), which contains a high nucleon density of $4.3 \times 10^{24}$ cm$^{-3}$. As shown in

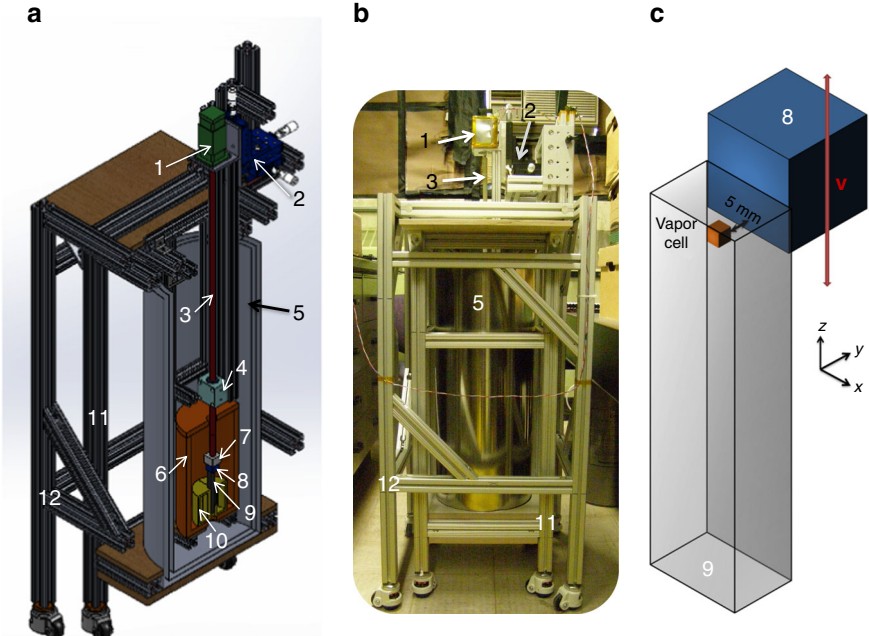

**Fig. 1** The experimental setup. **a** A cutaway schematic diagram of the experimental setup to probe the interaction $V$. **b** A photograph of the experimental setup. **c** An enlarged schematic of the configuration of a unpolarized bismuth germanate insulator (BGO) mass and a spin-exchange relaxation-free (SERF) atomic magnetometer (scaled). In **a–c**, 1 is a motor enclosed by a single-layer open $\mu$-metal box, 2 is a three-axis translation stage, 3 is a fiberglass G10 rod, 4 is a frictionless air bushing, 5 is a three-layer open $\mu$-metal co-axial cylindrical shield, 6 is a cylindrical ferrite shield, 7 is a sample holder, 8 is a unpolarized BGO mass, 9 is a SERF atomic magnetometer, and 10 is a plastic holder for the atomic magnetometer; 11 and 12 are aluminum frames holding the motor system moving the mass and the magnetometer/shields system, respectively. The frames are decoupled so as to reduce mechanical vibrations due to the mass motion. An unpolarized BGO mass is placed next to the rubidium atomic vapor cell located inside the head of the atomic magnetometer module. The BGO mass is translated up and down in $z$-direction with a constant velocity **v**

Fig. 1a, the BGO mass was connected to a stepper motor (LR43000 manufactured by Haydon Kerk Pittman), fixed on a three-axis translation stage (Thorlabs PT3), through a 1.5 m-long rigid fiberglass G10 cylindrical rod. The motor was positioned away from the magnetic shields and was enclosed by a single-layer open $\mu$-metal box to suppress magnetic noise arising from the motor operation [see Fig. 1b]. The translation stage precisely adjusted the position of the mass in the vicinity of the Rb vapor. In order to prevent the mass from oscillating due to the movement of the long G10 rod, a non-magnetic air bushing (OAV0500IB) was closely mounted to the ferrite shield, providing an additional support of the G10 rod with nearly frictionless motion of the mass. To reduce systematic effects due to vibrations induced by the motor moving the mass, the frame holding both the atomic magnetometer and the magnetic shields was decoupled from the frame holding the moving mass, as indicated in Fig. 1a, b.

As illustrated in Fig. 1c, the relative velocity term in the interaction $V$ was created by linearly moving the BGO mass along the $z$-axis with a constant velocity **v** next to the Rb vapor located in the magnetometer head. The standoff distance between the nearest surfaces of the mass and the vapor was set to 5 mm, limited by the location of the vapor. The Rb electron spins were oriented along the $x$-axis and the magnetometer was sensitive to the $z$ component of a magnetic field with the intrinsic field noise level of 15 fT Hz$^{-1/2}$, at low frequency. According to Eq. (3), the $\mathbf{B}_{\mathrm{eff}}$ is only along the $z$-axis in this configuration of the mass and the magnetometer because $\mathbf{B}_{\mathrm{eff}}$ is proportional to **v**.

Main sources of the systematic effects in this experiment are magnetic impurities buried inside the BGO mass generating fields measured on the order of $10^{-11}$ T and drift in the magnetometer signal on the order of $10^{-10}$ T. To suppress the systematic effects,

the mass movement was alternated to subtract the magnetometer signals between upward $v$ and downward $-v$ mass motions, where $v = |\mathbf{v}|$. This technique was highly effective since the sign of $\mathbf{B}_{\mathrm{eff}}$ is reversed for the opposite linear motions due to the single velocity term in Eq. (3), while the systematic effects are essentially the same[20].

In Fig. 2, we present the interaction potential, written as

$$P_{\mathrm{eff}} = \frac{\hbar}{8\pi}(2\widehat{\boldsymbol{\sigma}}_i \cdot \mathbf{v})\frac{e^{-r/\lambda}}{r}, \tag{4}$$

as a function of the distance between the center of the BGO mass and the center of the Rb vapor cell, defined as Shift here, at different interaction ranges. The interaction potential was numerically calculated by a Monte Carlo integration[20,21] which averaged the interaction potential $P_{\mathrm{eff}}$ over the both volumes of the mass and the vapor. This was done by randomly generating $2^{20}$ electron-nucleon pairs inside the volumes, summing the potential $P_{\mathrm{eff}}$ between each pair based on Eq. (4) by assuming the interaction range, and then normalizing the resulting potential for the nucleon density of the mass. More details are described below. The velocity magnitude, $v$, was 15.38 mm s$^{-1}$, used in the present experiments. As shown in Fig. 2, the interaction potential gets weaker as the mass moves away from the vapor and drops more drastically for the smaller interaction range due to the term of $e^{-r/\lambda}$ in Eq. (4). Since the interaction potential varies with the position of the mass with respect to the vapor, we selected one proper mass position near Shift = 0 in order to constrain the $g_A^e g_V^N$.

**Analysis.** In the signal subtraction method to suppress the systematic effects, the BGO mass was extended along the $z$ direction

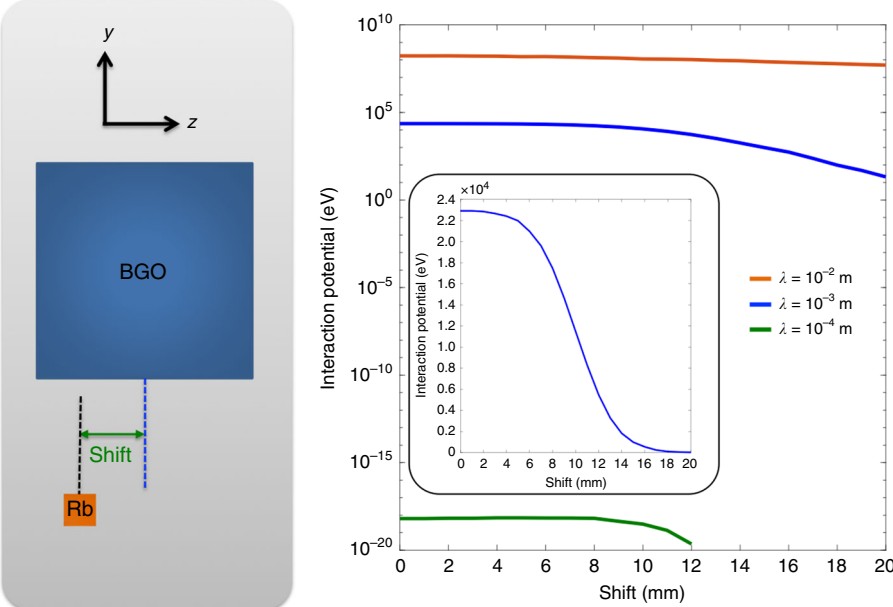

**Fig. 2** The interaction potential calculation. The interaction potential $P_{eff}$, numerically calculated by a Monte Carlo integration, as a function of the distance between the center of the bismuth germanate insulator (BGO) mass and the center of the rubidium (Rb) vapor cell, Shift, at different interaction ranges of $\lambda = 10^{-4}, 10^{-3}$, and $10^{-2}$ m. The vertical axis has a log scale. When the Shift = 0, the centers of the BGO mass and the Rb vapor cell are aligned. The inset enlarges the interaction potential at $\lambda = 10^{-3}$ m with the vertical axis in a linear scale

with the constant velocity of $v = 15.38$ mm s$^{-1}$ for 0.325 s from the initial Shift of $-1$ mm to the final Shift of 4 mm, and then it was retracted toward the initial configuration with $v = -15.38$ mm s$^{-1}$ for 0.325 s, as illustrated in Fig. 3a. The initial configuration was chosen because the sample holder contacted the head of the magnetometer when Shift $< -1$ mm. The motor was able to repeat this mass motion continuously, during which we recorded the magnetometer signals for 77 h using a 24-bit analog-to-digital converter (PXIe-4497 provided by National Instruments) with a sampling rate of 1 kHz.

Figure 3b shows standard time traces of the magnetometer signal, which present two full cycles of the mass motion reversal. Here one cycle refers to one extension with $v = 15.38$ mm s$^{-1}$ and one retraction with $v = -15.38$ mm s$^{-1}$ of the mass. The motor provided a voltage output which allows us to distinguish the mass motion's direction, as shown in Fig. 3c. This was also recorded simultaneously with the magnetometer signal. The rising and falling edges served as the reference points for each half cycle. The motor had a delay time of 0.05 s after each extension and retraction of the mass, necessary to reverse the mass motion's direction, thus the time elapsed for one cycle was 0.75 s. As discussed above, to extract the magnitude of the effective magnetic field, $B_{eff}$, from the magnetometer signals, we selected the middle configuration of each motion which corresponds to Shift = 1.5 mm, as shown in Fig. 3a, equivalent to the data point at the first 50% of the data in each half cycle, as shown in Fig. 3b. However, instead of taking only one data point in each half cycle for the extraction of $B_{eff}$, we used the mean of three data points, i.e., the data point at the first 50% of the data ± the adjacent data points. The distance between data points is 15 μm.

The mean value in each half cycle can be modeled as

$$D^{\pm}(t) = \pm B_{eff} + a_0 + a_1 t + a_2 t^2 + a_3 t^3, \quad (5)$$

where the sign of + and − corresponds to the sign of the velocity, $t$ is time, $a_0$ is the dc offset field on the order of a few hundred pT, as shown in Fig. 3b, and the last three terms characterize the slow drifts in the magnetometer signal expanded in a polynomial

function of time up to third order. These mainly arise from magnetometer electronics and temperature variations around the magnetometer. Note that $a_0$ remains constant as the mass linear motion is reversed, unlike $B_{eff}$. The magnetometer drift is mostly of first order: however, higher-order drifts can exist in the magnetometer signal due to, for example, opening/closing the laboratory door and other doors nearby the laboratory, or moving magnetic objects around the experimental setup. The simple difference between the first and the second half cycle with the time period of the half cycle, $T$, results in

$$D^{+}(T) - D^{-}(2T) = 2B_{eff} - a_1 T - 3a_2 T^2 - 7a_3 T^3. \quad (6)$$

Since this computation removes only the $a_0$ term, the remaining drift terms can add systematic effects to the extraction of $B_{eff}$. To this end, we used a more effective computation on the data which applies a $[+1 \ -3 \ +3 \ -1]$ weighting to the mean value from each half cycle within two cycles[20,40]:

$$D^{+}(T) - 3D^{-}(2T) + 3D^{+}(3T) - D^{-}(4T) = 8B_{eff} - 6a_3 T^3. \quad (7)$$

This weighting computation effectively eliminates the dominant slow drift terms up to second order. The remaining third- or higher-order drifts might become major systematic effects if the data is averaged for a long time and the statistical uncertainty is reduced. According to Eq. (7), the $B_{eff}$ can be obtained by dividing the resulting value by a factor of 8 assuming that the remaining third order drift is negligible.

A histogram of $B_{eff}$ extracted with the weighting computation from the magnetometer signals collected for 77 h, is shown in Fig. 4. A fit with a Gaussian distribution to the histogram gives $B_{eff} = (0.94 \pm 2.15) \times 10^{-16}$ T. According to Eq. (2), this equates to $\Delta E = (1.71 \pm 3.90) \times 10^{-20}$ eV with $\gamma = 2\pi \times 7.0 \times 10^9$ Hz T$^{-1}$[37]. The signal subtraction method together with the weighting computation suppresses the major systematic effects below the statistical sensitivity of $2.15 \times 10^{-16}$ T.

To experimentally constrain the interaction $V$ from our SERF magnetometer-based experiment, we performed a Monte Carlo

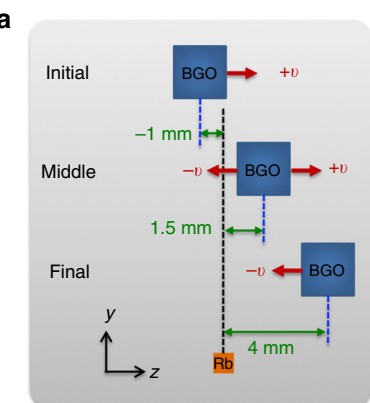

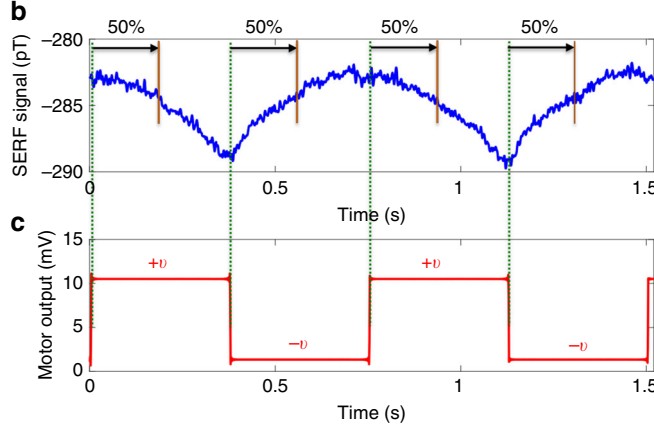

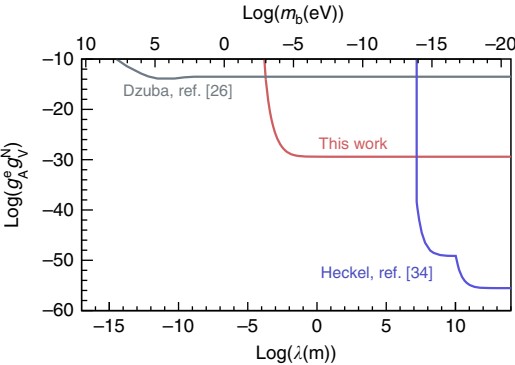

**Fig. 5** Experimental constraint on the interaction $V$. The red curve indicates the experimental limit of our experiment on the electron-nucleon coupling $g_A^e g_V^N$ with 1 $\sigma$ uncertainty as a function of the interaction range $\lambda$ in the bottom axis and the boson mass $m_b$ in the top axis. Our experiment is sensitive for $m_b < 10^{-3}$ eV and $\lambda > 10^{-4}$ m. The gray line indicates the same coupling which can be also derived from atomic parity non-conservation (PNC) experiments[26]. Our experiment significantly enhances the constraint for $g_A^e g_V^N$ by 17 orders of magnitude beyond the PNC experiments. The blue line indicates the constraint from the electron-spin polarized torsion pendulum[34]

integration for the interaction potential $P_{eff}$, as described above. Based on Eqs. (1) and (2), the experimental limit to $g_A^e g_V^N$ was estimated from dividing the experimental sensitivity of $\Delta E$ of $3.90 \times 10^{-20}$ eV by the interaction potential calculated at different interaction ranges[20]. Figure 5 shows the experimental limit to $g_A^e g_V^N$ in interaction ranges above $10^{-4}$ m (bottom axis), corresponding to the boson mass below $10^{-3}$ eV (top axis) with the data collection time of 77 h.

## Discussion

Our experiment sets an experimental limit on the parity-odd spin- and velocity-dependent interaction $V$ for electrons in the boson mass ranges below $10^{-3}$ eV, which is essential for developing directions in WISPs searches. One great advantage in our experiment is that the boson mass range can be simultaneously scanned, unlike the resonant cavity-based WISP-searching experiments, such as ADMX, which require tuning experimental parameters for each axion mass under investigation. Using this experimental method, recently, we also set an experimental limit on the parity-even spin- and velocity-dependent interaction $V_{4+5}$ between polarized electrons and unpolarized nucleons in the sub-meV range of the boson mass[20]. These experimental results demonstrate the feasibility of our experimental approach using the SERF atomic magnetometer. Our experiments will play an important role in exploring the uninvestigated spin-dependent interactions for polarized electrons[21].

Further enhancement in the experimental sensitivity could be achievable. The SERF atomic magnetometer used in this work operates in a single-beam configuration in which a single circularly polarized laser beam is used to pump and probe the Rb electron spins[39]. The single-beam configuration is suboptimal in terms of magnetic field sensitivity. For an improved design, the magnetometer would use an orthogonal-beam configuration in which the probe beam is perpendicular to the pump beam, and the laser beams' wavelengths are individually optimized. This modification can improve the sensitivity to 1 fT Hz$^{-1/2}$ immediately. Further work will bring the magnetometer close to the fundamental photon shot noise limit of subfemto-Tesla to atto-Tesla[41] by reducing the noise sources from laser frequency and intensity fluctuations and by using the hybrid optical pumping[41].

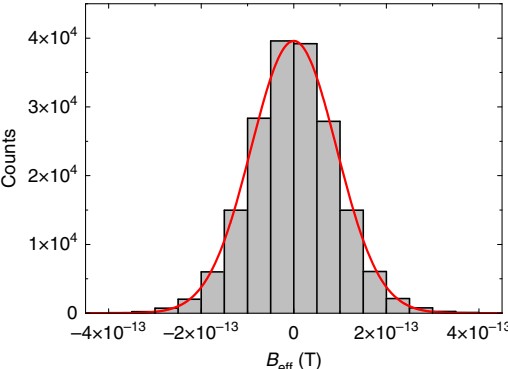

**Fig. 3** Data collection. **a** A repeated bismuth germanate insulator (BGO) mass linear motion with respect to the rubidium vapor (not scaled). The BGO mass was extended with $v = 15.38$ mm s$^{-1}$ for 0.325 s from the initial configuration to the final configuration and then retracted toward the initial configuration with $v = -15.38$ mm s$^{-1}$ for 0.325 s, by a motor. **b** Time traces of spin-exchange relaxation-free (SERF) magnetometer signal showing two full cycles of the mass linear motion reversal. **c** The voltage output from the motor indicating the motion's direction which was high and low at extraction and retraction of the mass. The rising and falling edges served as the reference points for each half cycle

**Fig. 4** Histogram of the magnitude of the effective field, $B_{eff}$. Distribution of $B_{eff}$ is obtained with the weighting computation from data collected for 77 h. The overlaid solid curve represents a fit with a Gaussian distribution to the data, which gives $B_{eff} = (0.94 \pm 2.15) \times 10^{-16}$ T. The major systematic effects are below the statistical sensitivity of $2.15 \times 10^{-16}$ T

In conclusion, we explored the exotic parity-odd spin- and velocity-dependent interaction $V$ for polarized electrons based on an experimental approach utilizing a SERF atomic magnetometer. Our experiment sets an experimental limit on the interaction in the interaction range higher than $10^{-4}$ m corresponding to the boson mass lower than $10^{-3}$ eV.

## Methods

**Monte Carlo integration**. The algorithm of the Monte Carlo integral is following:

(1) $2^{20}$ random pairs of points inside both the volumes of the BGO mass and the Rb vapor cell are generated;

(2) An interaction range $\lambda$ between $10^{14}$ to $10^{-16}$ m is assumed to calculate the interaction potential $P_{\text{eff}}^i$ between a randomly generated pair of points $i$,

$$P_{\text{eff}}^i = \frac{\hbar}{8\pi}(2\hat{\boldsymbol{\sigma}}_i \cdot \mathbf{v})\frac{e^{-r_i/\lambda}}{r_i};\qquad(8)$$

(3) All the contributions to the potential are summed and normalized to give the average interaction potential for the nucleon density of BGO mass $N_{\text{BGO}}$:

$$P_{\text{eff}} = N_{\text{BGO}}\frac{1}{2^{20}}\sum_i^{2^{20}} P_{\text{eff}}^i;\qquad(9)$$

(4) The experimental limit to the coupling strength $g_A^e g_V^N$ is derived by dividing the experimental sensitivity of the energy shift $\Delta E$ for Rb atoms by the calculated average interaction potential;

(5) The steps from (1) to (4) are repeated with another given interaction range between $10^{-1}$ to $10^{-6}$ m.

The Monte Carlo error can be ignored because the error numerically decreases as $1/\sqrt{N}$ where $N$ is the number of random points, thus it is less than 1% with the $2^{20}$ random points.

**SERF atomic magnetometer**. A SERF atomic magnetometer employed for the experiments, manufactured by QuSpin Inc., is a compact, self-contained unit with all the necessary optical components and can be readily operated using a control software. Practically, the SERF regime is achieved by operating in a near-zero magnetic field and by heating the Rb vapor; a SERF magnetometer is sensitive at low frequency below 100 Hz. It is mainly composed of a 3 mm cube $^{87}$Rb vapor cell, a single semiconductor 795 nm laser for both optical pumping and probing, and silicon photodiodes[39]. The dimensions of the magnetometer module are: length 8 cm, width 1.4 cm, and height 2.1 cm. The cell temperature is around 160 °C for sufficiently large Rb density. The action of the laser beam creates a large number of polarized Rb electron spins in the vapor cell and its intensity transmitted through the cell is detected using the photodiode. The output from the photodoide is an analogue voltage signal and it can be converted to a magnetic field signal by applying a known calibration field to the vapor cell. In order to reduce the effects of the laser noise due to intensity fluctuation, the magnetometer uses a modulating field at 926 Hz[39]. The measured field at a certain low frequency is restored using a built-in lock-in amplifier and a low-pass filter.

## Data availability

The data that support the findings of this study are available from the corresponding author upon reasonable request.

## Code availability

The code of the Monte Carlo integral is available from the corresponding author upon reasonable request.

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

## Acknowledgements

The authors gratefully acknowledge this work was supported by the Los Alamos National Laboratory LDRD office through grant 20180129ER.

## Author contributions

Y.J.K. designed and carried out the experiments, and analyzed the data. P.-H.C. performed the Monte-Carlo simulation and estimated the sensitivity. I. S. provided expertise in atomic magnetometers. S.N. and Y.J.K. built the experimental setup. All authors contributed to the manuscript.

## Additional information

**Competing interests:** The authors declare no competing interests.

