## [Peer Review File · Nature Communications]

EDITORIAL NOTE: Images have been redacted from reports due to third party rights.

Reviewers' comments:

Reviewer #1 (Remarks to the Author):

This manuscript describes a nice result. The authors use a highly sensitive atomic magnetometer to probe a parity-violating spin-velocity interaction that could arise from the exchange of exotic spin 1 bosons between the polarized electrons in their magnetometer and a moving source. The result is surprising because the source velocity they achieved is a tiny fraction (50 parts per trillion) of the speed of light and yet they were able to improve on an existing constraint from atomic parity-violation experiments by up to 17 orders of magnitude for a restricted range of possible boson masses.

The readability and accessibility of the paper would be much improved if THROUGHOUT the authors use the preferred notation of their reference 3, replacing f_{12+13} with the more basic and physical $g^e_A g^N_V$ and everywhere eliminating the subscript "12+13". They also should use the symbol m_b rather than m_a which could be confused for axion mass.

This starts in equation 1 where f_{12+13} should be dropped on both sides of the equality sign and on the right side replaced with $g^e_A g^N_V$. For example V_{12+13} becomes simply V , B_{12+13} becomes B_{eff} , etc. My strongly suggested change in notation affects every equation and figures 4 and 5 as well as numerous places in the text. The caption of Figure 5 contains the statement that " $f_{12+13}=4 g^e_A g^N_V$ " citing ref[31]. I suggest that the authors contact one of authors of their Ref 3 to check on factors of 2 before they submit a revised manuscript.

Reviewer #2 (Remarks to the Author):

Manuscript NCOMMS-19-03484 reports new bounds on an exotic velocity-dependent interaction between electrons and nucleons from their atomic magnetometry experiment. The reported limits improve on previous bounds by up to 17 orders of magnitude, which is very impressive. The manuscript is well-written, the details are explained clearly, and the main results and conclusions appear to be correct. I think the work is of broad interest to the physics community, as it spans several different fields of physics, including atomic physics and particle physics. I recommend publication in Nature Communications after some relatively minor suggestions and modifications are taken into account (below).

I was not able to find the details in Ref. [31] for why there is the numerical coefficient 4 in the relation $f_{12+13} = 4 g^e_A g^N_V$. Obviously, the numerical coefficient will depend on the precise definition of the interaction Lagrangian for the vector boson. However, for the "standard" interaction Lagrangian for the vector boson given in Refs. [2] and [3], one instead finds $f_{12+13} = 1 g^e_A g^N_V$. While we are on the topic of symbols and notation, is there a reason to change from the use of the notation $g^e_A g^N_V$ at the start of the paper to the notation f_{12+13} later on? In my opinion, an unfamiliar reader may find the latter notation confusing. Why not just stick to the clearer notation $g^e_A g^N_V$ throughout?

It would be reasonable, and indeed recommended, to extend the main results graph in Fig. 5 to also show the smaller axion masses (larger interaction ranges). I think it would be particularly helpful for the reader to also see the torsion-pendulum constraints from Refs. [36] and [37] included in this graph. This way the reader could see the limits on $g^e_A g^N_V$ arising from the three different types of experiments which to the best of my knowledge constitute all of the currently known experimental limits on $g^e_A g^N_V$.

1. Reviewer #1 (Remarks to the Author):

This manuscript describes a nice result. The authors use a highly sensitive atomic magnetometer to probe a parity-violating spin-velocity interaction that could arise from the exchange of exotic spin-1 bosons between the polarized electrons in their magnetometer and a moving source. The result is surprising because the source velocity they achieved is a tiny fraction (50 parts per trillion) of the speed of light and yet they were able to improve on an existing constraint from atomic parity-violation experiments by up to 17 orders of magnitude for a restricted range of possible boson masses.

The readability and accessibility of the paper would be much improved if THROUGHOUT the authors use the preferred notation of their reference 3, replacing f_{12+13} with the more basic and physical $g_A^e g_V^N$ and everywhere eliminating the subscript "12+13". They also should use the symbol m_b rather than m_a which could be confused for axion mass. This starts in equation 1 where f_{12+13} should be dropped on both sides of the equality sign and on the right side replaced with $g_A^e g_V^N$. For example V_{12+13} becomes simply V , B_{12+13} becomes B_{eff} , etc. My strongly suggested change in notation affects every equation and figures 4 and 5 as well as numerous places in the text.

We modified the notations from f_{12+13} to $g_A^e g_V^N$, from m_a to m_b , from V_{12+13} to V , from B_{12+13} to B_{eff} , and from P_{12+13} to P_{eff} throughout the manuscript.

The caption of Figure 5 contains the statement that " $f_{12+13} = 4g_A^e g_V^N$ " citing ref[31]. I suggest that the authors contact one of authors of their Ref 3 to check on factors of 2 before they submit a revised manuscript.

We appreciate that the reviewer pointed out the mistake. According to Eq. 3 in Ref. [31], assuming $f_v^{ep} = f_v^{en} = f_v^{eN}$, and neglecting f_v^{ee} ,

$$A f_{12+13} = Z f_v^{ep} + (A - Z) f_v^{en} = A f_v^{eN}. \quad (1)$$

Considering the Table I in Ref. [31], assuming $f_v^{ep} = f_v^{en} = f_v^{eN}$ and $g_V^p = g_V^n = g_V^N$, and neglecting f_v^{ee} and g_V^e ,

$$f_v^{ee} + f_v^{ep} + f_v^{en} = 2g_A^e [2g_V^e + g_V^p + g_V^n] \quad (2)$$

$$f_v^{ep} + f_v^{en} = 2f_v^{eN} = 4g_A^e g_V^N. \quad (3)$$

Additionally, in Ref. [2], Eq. 4.3 shows

$$V^{en}(\vec{r}, \vec{v}) = \frac{1}{8\pi r} f_v^{en} \vec{\sigma} \cdot \vec{v} e^{-m_0 r} \quad (4)$$

where $m_0 = 1\lambda$ and $f_v^{eN} \sim 2g_A^e g_V^N$ as shown in Eq. 5.28

$$f_v^{eN} = 2\left(1 + \frac{m_e}{m_N}\right) g_A^e g_V^N. \quad (5)$$

Therefore, $f_{12+13} = f_v^{eN} = 2g_A^e g_V^N$. We updated the result from $f_{12+13} = 4g_A^e g_V^N$ to $f_{12+13} = 2g_A^e g_V^N$ accordingly.

2. Reviewer #2 (Remarks to the Author):

Manuscript NCOMMS-19-03484 reports new bounds on an exotic velocity-dependent interaction between electrons and nucleons from their atomic magnetometry experiment. The reported limits improve on previous bounds by up to 17 orders of magnitude, which is very impressive. The manuscript is well-written, the details are explained clearly, and the main results and conclusions appear to be correct. I think the work is of broad interest to the physics community, as it spans several different fields of physics, including atomic physics and particle physics. I recommend publication in Nature Communications after some relatively minor suggestions and modifications are taken into account (below).

I was not able to find the details in Ref. [31] for why there is the numerical coefficient 4 in the relation $f_{12+13} = 4g_A^e g_V^N$. Obviously, the numerical coefficient will depend on the precise definition of the interaction Lagrangian for the vector boson. However, for the standard interaction Lagrangian for the vector boson given in Refs. [2] and [3], one instead finds $f_{12+13} = 1g_A^e g_V^N$. While we are on the topic of symbols and notation, is there a reason to change from the use of the notation $g_A^e g_V^N$ at the start of the paper to the notation f_{12+13} later on? In my opinion, an unfamiliar reader may find the latter notation confusing. Why not just stick to the clearer notation $g_A^e g_V^N$ throughout?

Thank you for the suggestion. Reviewer 1 also mentioned this question. We updated to $f_{12+13} = 2g_A^e g_V^N$ as discussed above.

It would be reasonable, and indeed recommended, to extend the main results graph in Fig. 5 to also show the smaller axion masses (larger interaction ranges). I think it would be particularly

helpful for the reader to also see the torsion-pendulum constraints from Refs. [36] and [37] included in this graph. This way the reader could see the limits on $g_A^e g_V^N$ arising from the three different types of experiments which to the best of my knowledge constitute all of the currently known experimental limits on $g_A^e g_V^N$.

Thank you for the suggestion. We updated Figure 5 accordingly.

All modifications in the manuscript

- Update Ref. [3] from arXiv:1810.10364 to Phys. Rev. A 99, 022113 (2019).
- Update Ref. [30] from arXiv:1710.02504 to Rev. Mod. Phys. 91,015001 (2019).
- Correct the volume of Ref. [13] and delete arXiv:1201.5902 [hep-ph].
- Change all f_{12+13} to $2g_A^e g_V^N$.
- Change all V_{12+13} to V and all m_a to m_b .
- Update Eq. 1 to $V(\mathbf{e}_{\sigma_i}, \mathbf{r}) = g_A^e g_V^N \frac{\hbar}{4\pi} (\mathbf{e}_{\sigma_i} \cdot \mathbf{v}) \frac{e^{-r/\lambda}}{r}$.
- Modify to “The coupling strength constant $g_A^e g_V^N$ is between electrons and nucleons for the interaction assuming no difference between electron-neutron and electron-proton couplings, and ignoring electron-electron couplings” in the first paragraph in page 4.
- Delete “where the coupling has been scaled according to $f_{12+13} = 4g_A^e g_V^N$ [31]” in the subtitle of Fig. 5
- Change all B_{12+13} to B_{eff} and all P_{12+13} to P_{eff} .
- Change the vertical axis of Fig. 5 from f_{12+13} to $g_A^e g_V^N$.
- Update the range to larger λ in Fig. 5 and include the torsion-pendulum constraint.
- Add “The blue line indicates the constraint from the torsion pendulum [37]” in the subtitle of Fig. 5.
- Add a code availability statement in the Methods section to ensure compliance with Nature Research editorial policies.

- Slightly modify the Abstract to meet the required length, no more than 150 words.
- Express all unit dimensions using negative integers, e.g., from $\text{fT}/\text{Hz}^{1/2}$ to $\text{fT Hz}^{-1/2}$, and all vectors in bold without italics to meet the Nature Research editorial policies.
- Modify to “ \mathbf{r} is the separation vector in the direction between the polarized electron and the unpolarized nucleon, $r = |\mathbf{r}|$ ” in the last paragraph in page 3.
- Modify to “between upward v and downward $-v$ mass motions, where $v = |\mathbf{v}|$ ” in page 6.
- Correct “beyond the SM” to “beyond the Standard Model of particle physics” in page 3.
- Modify to “An interaction range λ between 10^{14} to 10^{-16} m” in the Monte Carlo integration Subsection in Methods, due to the update of the interaction range to larger λ in Fig. 5.

Reviewers' comments:

Reviewer #1 (Remarks to the Author):

Reviewer's Report on revised manuscript by Y.J. Kim et al.

Since my review of the original manuscript and after reading the revised manuscript I have become more concerned about the suitability of this manuscript for Nature Communications.

The importance of the work is more modest than the manuscript would lead one to think. I suspect that one can obtain an even tighter limit on $g_A^e g_V^N$ than the one obtained in this work by using a procedure similar to that employed in Ref. 34: ie. getting g_A^e and g_V^N from other existing experimental results.

Furthermore, text is not up to Nature Communications' standards and could be considerably condensed as there is a fair amount repetition. The introduction in particular is not well organized and the Results section is clear but perhaps goes into more detail than needed. The paper could easily be shortened by a factor of 2 with no loss of relevant information.

The key Figure 5 which shows the results needs to be improved.

1) The positions of the tick labels on the horizontal axes are ambiguous. This could be solved by treating the horizontal axis like the vertical axis which is unambiguous.

2) The Figure caption does not state the confidence level of the constraints. On the other hand, it repeats much information already given in the text.

The final sentence of the abstract should be changed from "...the electron-nucleon coupling strength of ..." to the electron-nucleon coupling $g_A^e g_V^N < \dots$

I urge the authors to employ EVERYWHERE the notation of Ref.3, the most recent and definitive work on the subject of spin dependent forces, and omit their Ref. 31. For example, consider Eqs. 1 and 3. According to Ref. 3 the denominator 4π should be 8π . If the author's constraint is based on Eqs. 1 and 3 summed over the nucleons in the source as stated, then their constraint is actually off by a factor of 2. By the way, one normally denotes an electron unit polarization as σ_e and not e_σ

The ADMX references are out of date, Refs. 15 and 16 should be replaced Du et al PRL 120 (2018). In three places "torsion pendulum" would be clearer if it read "electron-spin polarized torsion pendulum".

In summary, this paper describes a nice piece of work where, by using a sensitive atomic magnetometer, a moving a source with a velocity of only 15 mm/s can make a sensitive probe of v/c effects. This is impressive! But the paper is not well written and the resulting limit on $g_A^e g_V^N$ could well be more than equaled by combining results from existing experiments.

Reviewer #2 (Remarks to the Author):

I believe that the main issues raised by myself and the other reviewer have been addressed and that the manuscript can now be published.

1. Reviewer #1 (Remarks to the Author): Reviewers Report on revised manuscript by Y.J. Kim et al.

Since my review of the original manuscript and after reading the revised manuscript I have become more concerned about the suitability of this manuscript for Nature Communications. The importance of the work is more modest than the manuscript would lead one to think. I suspect that one can obtain an even tighter limit on $g_A^e g_V^N$ than the one obtained in this work by using a procedure similar to that employed in Ref. 34: ie. getting g_A^e and g_V^N from other existing experimental results.

Thanks for the reviewer's suggestion. In order to obtain the constraint by combining different experiments, the process is quite complicated and may be inaccurate. First we can obtain $(g_A^e)^2$ from Fig. 18 in Rev. Mod. Phys **90**, 025008 (2018). Second we can obtain $(g_A^n g_V^N)$ from Fig. 1 in Phys. Rev. D **88**, 031101(R) (2013) and Fig. 2 in Phys. Rev. Lett. **110**, 082003 (2013). Third we can obtain $(g_A^n g_A^n)$ from Fig. 5.12(a) on page 207 in G. Vasilakis, Ph.D. thesis, Princeton University, 2007.

The constraint of g_A^n is about $g_A^n < 3.46 \times 10^{-21}$ from the square root of $(g_A^n)^2$ in Fig. 5.12(a) on page 207 in G. Vasilakis, Ph.D. thesis, Princeton University, 2007 for $\lambda > \sim 10^{-9}m$. By dividing $(g_A^n g_V^N)$ by g_A^n , we can obtain the constraint of g_V^N . We can obtain the constraint of g_A^e from the square root of $(g_A^e)^2$. Figure 1 shows the constraint of g_V^N and g_A^e separately. By multiplying two constraints, in principle, we can obtain the constraint of $g_A^e g_V^N$. Figure 2 shows the constraints of $g_A^e g_V^N$ together with existing direct measurements.

Although we could obtain the same constraint by combining other experiments, the process includes combination of very different experiments and the uncertainty propagation is not straightforward. Direct measurement of this interaction is still valuable. We prefer not to add this constraint by combining experiments in the manuscript but only mention it in this response.

Figure 1: The constraint of g_V^N and g_A^e .

Figure 2: The constraint of $g_A^e g_V^N$.

Furthermore, text is not up to Nature Communications standards and could be considerably condensed as there is a fair amount repetition. The introduction in particular is not well organized and the Results section is clear but perhaps goes into more detail than needed. The paper could easily be shortened by a factor of 2 with no loss of relevant information.

Thank you for the comment. We re-organized the Introduction, and we condensed the Introduction as well as the Results by deleting some repetitions and details.

The key Figure 5 which shows the results needs to be improved. 1) The positions of the tick labels on the horizontal axes are ambiguous. This could be solved by treating the horizontal axis like the vertical axis which is unambiguous. 2) The Figure caption does not state the confidence level of the constraints. On the other hand, it repeats much information already given in the text.

We changed the x -axis (top and bottom axes) to a log scale as the y -axis shown in Fig. 3. And we deleted some information in the figure caption, already given in the text. Our limit is calculated with 1σ uncertainty. We also added this in the figure caption.

Figure 3: Experimental constraint on the interaction V .

The final sentence of the abstract should be changed from “...the electron-nucleon coupling strength of ...” to the electron-nucleon coupling $g_A^e g_V^N < \dots$ ”

We modified the final sentence to “We set an experimental limit on the electron-nucleon cou-

pling $g_A^e g_V^N < 10^{-30}$ at the mediator boson mass below 10^{-4} eV, significantly improving the current limit by up to 17 orders of magnitude”.

I urge the authors to employ EVERYWHERE the notation of Ref.3, the most recent and definitive work on the subject of spin dependent forces, and omit their Ref. 31. For example, consider Eqs. 1 and 3. According to Ref. 3 the denominator 4π should be 8π . If the authors constraint is based on Eqs. 1 and 3 summed over the nucleons in the source as stated, then their constraint is actually off by a factor of 2. By the way, one normally denotes an electron unit polarization as σ_e and not e_σ

We replaced e_{σ_i} with $\hat{\sigma}_i$. We also employed the notation of Ref. 3, which leads to the use of 8π instead of 4π in Eqs.1, 3, and 4 and the equation in the Methods:

$$V(\hat{\sigma}_i, \mathbf{r}) = g_A^e g_V^N \frac{\hbar}{8\pi} (2\hat{\sigma}_i \cdot \mathbf{v}) \frac{e^{-r/\lambda}}{r}, \quad (1)$$

$$\mathbf{B}_{\text{eff}} = g_A^e g_V^N \frac{2\mathbf{v}}{8\pi\gamma} \frac{e^{-r/\lambda}}{r}, \quad (2)$$

and

$$P_{\text{eff}} = \frac{\hbar}{8\pi} (2\hat{\sigma}_i \cdot \mathbf{v}) \frac{e^{-r/\lambda}}{r}. \quad (3)$$

Equation 8 in Ref. [3],

$$V_{AV} = g_1^A g_2^V \boldsymbol{\sigma}_1 \cdot \left\{ \frac{\mathbf{p}_1}{m_1} - \frac{\mathbf{p}_2}{m_2}, \frac{e^{-Mr}}{8\pi r} \right\} - \frac{g_1^A g_2^V}{2} (\boldsymbol{\sigma}_1 \times \boldsymbol{\sigma}_2) \cdot \hat{\mathbf{r}} \frac{e^{-Mr}}{4\pi m_2}, \quad (4)$$

can be simplified as

$$V_{AV} = g_1^A g_2^V \boldsymbol{\sigma}_1 \cdot \left\{ \mathbf{v}, \frac{e^{-Mr}}{8\pi r} \right\} = g_1^A g_2^V \boldsymbol{\sigma}_1 \cdot \frac{2\mathbf{v}e^{-Mr}}{8\pi r} \quad (5)$$

for $\boldsymbol{\sigma}_2 = 0$ and the relative velocity $\mathbf{v} = \frac{\mathbf{p}_1}{m_1} - \frac{\mathbf{p}_2}{m_2}$. This is identical to Eq. 1 in the manuscript. We replaced Ref. [31] with Ref. [3].

The ADMX references are out of date, Refs. 15 and 16 should be replaced Du et al PRL 120 (2018). In three places torsion pendulum would be clearer if it read electron-spin polarized torsion pendulum.

Thank you for the comment. We replaced Refs. 15 and 16 with Phys. Rev. Lett. 120, 151301 (2018). We also replaced “torsion pendulum” with “electron-spin polarized torsion pendulum”.

In summary, this paper describes a nice piece of work where, by using a sensitive atomic magnetometer, a moving a source with a velocity of only 15 mm/s can make a sensitive probe of v/c effects. This is impressive! But the paper is not well written and the resulting limit on $g_e^A g_N^V$ could well be more than equaled by combining results from existing experiments.

We updated the manuscript as described above.

2. Reviewer #2 (Remarks to the Author):

I believe that the main issues raised by myself and the other reviewer have been addressed and that the manuscript can now be published.

We appreciate the reviewer's positive feedback.

REVIEWERS' COMMENTS:

Reviewer #1 (Remarks to the Author):

The authors have done a good job of improving the readability of their paper. It should be published.

REVIEWERS' COMMENTS:

Reviewer #1 (Remarks to the Author):

The authors have done a good job of improving the readability of their paper. It should be published.

We appreciate the reviewer's positive feedback.